# Flexible and Highly Sensitive Strain Sensor Based on Laser-Induced Graphene Pattern Fabricated by 355 nm Pulsed Laser

**DOI:** 10.3390/s19224867

**Published:** 2019-11-08

**Authors:** Sung-Yeob Jeong, Yong-Won MA, Jun-Uk Lee, Gyeong-Ju Je, Bo-sung Shin

**Affiliations:** 1Interdisciplinary Department for Advanced Innovative Manufacturing Engineering, Pusan National University, Pusan 46241, Korea; ysjsykj8025@naver.com (S.-Y.J.); decentsoul@pusan.ac.kr (Y.-W.M.); 2Department of Optics and Mechatronics Engineering, Pusan National University, Pusan 46241, Korea; lju3534@naver.com (J.-U.L.); ysjsykj8025@pusan.ac.kr (G.-J.J.)

**Keywords:** laser-induced graphene (LIG), 355 nm pulsed laser, strain sensor, polyimide, polydimethylsiloxane (PDMS)

## Abstract

A laser-induced-graphene (LIG) pattern fabricated using a 355 nm pulsed laser was applied to a strain sensor. Structural analysis and functional evaluation of the LIG strain sensor were performed by Raman spectroscopy, scanning electron microscopy (SEM) imaging, and electrical–mechanical coupled testing. The electrical characteristics of the sensor with respect to laser fluence and focal length were evaluated. The sensor responded sensitively to small deformations, had a high gauge factor of ~160, and underwent mechanical fracture at 30% tensile strain. In addition, we have applied the LIG sensor, which has high sensitivity, a simple manufacturing process, and good durability, to human finger motion monitoring.

## 1. Introduction

Recently, metal wires, carbon nanotubes, and graphene have been applied to substrates such as polyethyleneimine (PEI), polyethylene terephthalate (PET), ECOFLEX, and polydimethylsiloxane (PDMS); the resulting materials are mechanically flexible and robust, and are used in many areas such as touch screens, biomedical devices, wearable devices, human–robot interfaces, solar cells, and supercapacitors [1,2,3,4,5,6,7,8,9,10,11,12,13,14,15]. In particular, graphene is widely used in sensors due to its high electrical conductivity and various superior mechanical characteristics [7,11,16,17,18,19,20]. Graphene production methods include (1) exfoliation from graphite crystals and (2) chemical vapor deposition (CVD) involving deposition of carbon on a transition metal at high temperature [21,22]. Recently, research on the production of porous graphene using lasers, i.e., laser-induced graphene (LIG), has been reported [21,22]. Irradiation of a laser onto polyimide film at a certain degree of laser fluence, i.e., the optical energy delivered per unit area, resulted in photothermal ablation by the locally high temperature, which breaks the chemical bonds of the polyimide film; this process has been reported to produce LIG with a porous multilayer structure [11,20,21,22]. The LIG has improved electrical properties due to the conversion of sp3 carbon atoms to sp2 carbon atoms. [11]. Therefore, LIG is widely applied to humidity-, bio-, and mechanical sensors [11,18,19,22]. Mechanical sensors include strain sensors and pressure sensors; PDMS, which is actively used as substrate for these sensors, is thermally and chemically stable, physically robust, and flexible, with a shear modulus of 250 kPa, loss tangent <<0.001, and Young’s modulus of 750 kPa [14,15]. In addition, PDMS is transparent, nonfluorescent, nontoxic, and biocompatible; it undergoes no adhesion with polymer films, and so is suitable for use in MEMS and biomedical applications [14,15].

Strain sensors are generally sensors in which an electrical signal changes according to strain, indicating the degree of deformation of an object according to the applied force. Conventional metal-based strain sensors have a low fabrication cost but also a low gauge factor (GF) sensitivity of ~2 and maximum measurable strain of ~5% [4,5,6,7,8,9,10]. Therefore, other strain sensors have been reported that have a piezoresistive effect that overcomes the shortcomings of conventional sensors; these sensors combine an elastomer and a material with a multidimensional structure, such as graphene or carbon nanotubes (CNTs). Amjadi et al. reported a strain sensor based on silver nanowire–elastomer nanocomposite; it showed strong piezoresistivity, with a GF in the range of 2–14, and maximum stretchability of 70%. [4]. Ryu et al. reported a CNT composite sensor and demonstrated its good sensitivity (GF~117) [5]. Bae et al. reported a graphene-based sensor that had piezoresistive properties that were present under strains up to 7.1% [7]. Stretchable CNT strain sensors [6,9,12,13,19,23] and ZnO wire sensors (GF~1250) [8] have also been reported. Despite many attempts to fabricate strain sensors that use sensitive, high-performance nanoparticles, such as CNT and graphene, previous research on sensors has led to good sensitivity and mechanical performance but was unsuccessful in making the fabrication process easier or in reducing the fabrication costs. Therefore, a one-step direct laser writing (DLW) method is proposed to simplify the process and reduce fabrication costs [18].

In this paper, we fabricate an LIG pattern using a 355 nm pulsed laser; we then implant the pattern on an elastomeric substrate, suggesting the possibility of its application as a strain sensor. To simplify the fabrication process, patterns are fabricated using the DLW method, and the electrical characteristics such as electrical resistance and sheet resistance are evaluated. In addition, structural analysis of the LIG pattern according to the laser fluence and laser focal length, as well as evaluation of device applicability as a sensor, is performed.

## 2. Experimental

### 2.1. Fabrication of LIG Pattern

We used a 355 nm UV pulsed laser to form LIG on a polyimide film (DuPont™ Kapton® HN); we then used the film as a strain sensor. The 355 nm pulsed laser is widely used for microfabrication. The laser setup and specifications are shown in Figure 1 and Table 1, respectively.

Polyimide has 90% absorption for the 355 nm wavelength and this absorption increases with increasing laser fluence [24]. When the 355 nm pulsed laser is irradiated onto polyimide film, the surface temperature rises locally to over 1000 °C, which causes ablation by photothermal phenomenon [19]. The polyimide film releases oxygen and nitrogen gas at 550 °C and is carbonized at 700 °C; the carbonized polyimide forms a porous graphitic structure [22]. Figure 2 shows the generation principle of the LIG. Since the C–N binding force of polyimide is the lowest, it is the first to be broken and separated. In addition, separated unstable compounds exist in various forms, such as CO, CN, C, C_2_, CH, C_2_H_2_, HCN, in combination with each other or other elements in the atmosphere [25].

In order to observe structural changes of the LIG pattern, the laser was irradiated onto the substrate at different laser fluences; the laser fluence was set by adjusting the laser power and the scanning speed. A 25 μm-thick polyimide film was used to better implant the LIG pattern in PDMS; the degrees of implantation and carbonization were evaluated according to laser fluence.

### 2.2. Fabrication of Flexible and Bendable Strain Sensor

In order to improve the sensitivity of the sensor and to obtain a uniformly thin thickness of the substrate and top, spin coating, a common microfabrication method for polymer films, was used for fabrication of PDMS. The PDMS was spin-coated at 100 rpm for 5 min, leading to a uniform thickness of 400 μm. The sensor fabrication process is shown in Figure 3a.

By placing a PI film on the PDMS and irradiating the 355 nm pulsed laser with laser fluence under certain conditions, the desired graphitic pattern was implanted on the substrate by the Galvano scanner (Figure 1). When the remaining polyimide film on the PDMS was removed, only the LIG pattern was implanted on the PDMS, which was then covered with PDMS to protect the pattern. This demonstrates the fabrication of sensitive and flexible sensors in a simple and inexpensive process without any technology, overcoming the disadvantages of previously reported piezoresistive strain sensors [4,5,6,7,8,9,10,12,13,18,26,27].

Strain sensors are evaluated according to their sensitivity, maximum strain, response speed, stability, and fabrication cost. In particular, the GF, which is expressed as the change in the electrical resistance of the strain to the sensitivity of the strain sensor, can be used as a typical measure of efficiency.
(1)$$\mathrm{Gauge}\;\mathrm{Factor}=\;\frac{\Delta R/R_{0}}{\varepsilon }$$
where ΔR is the variation of electrical resistance when strain is applied, R_0_ is the initial electrical resistance, and ε is the strain.

The LIG strain sensor in this research measured strain through bending and tension. Strain generations by bending and elongation are shown in Figure 3b. The change of electrical resistance to strain was measured using a sourcemeter (Keithley®, sourcemeter 2450).

Figure 3c shows the electrical properties of the carbonized pattern depending on the laser fluence: the electrical resistance value ranged from 237 Ω to 60 kΩ and sheet resistance ranged from 10.86 to 65 Ω/□.

## 3. Results and Discussion

### 3.1. Morphological Characterization of the LIG Pattern

The carbonization of the LIG pattern is mainly due to photothermal ablation [20,28,29]; the degree of carbonization depends on laser parameters such as laser power, scanning speed, and focal length. We set up various laser fluences to produce a porous LIG pattern, which is shown in Figure 4. The LIG pattern was fabricated by the DLW method; the morphology of the pattern was observed by field emission scanning microscopy (FE-SEM). Samples were platinum coated before observation.

The SEM images in Figure 4a–d show the graphene-like and porous structure of several microscale graphene-like flakes; the line width of the LIG pattern is about 30 μm and the sensor has a biaxial strain gauge. Lines fabricated by the DLW method can form very thin and desirable shape patterns, suggesting that strain gauges grow rapidly when the laser scanning direction and elongation or bending direction are the same. The graphene-like flakes can be seen forming fine graphitic projections of several micro-meters. The LIG pattern fabricated by the DLW method under various laser irradiation parameters usually has a line width in the range of approximately several tens of micrometers. The influence of laser fluence on the structure and morphology of the material was studied by irradiating the laser at a laser fluence from 5.6 to 7.4 J/cm^2^. When the laser is irradiated beyond the specific fluence that can produce graphene, the characteristics of the porous structure gradually deteriorate, the permittivity is lowered, and the electrical resistance becomes smaller, but we have found the optimal laser fluence creates graphene-like flake patterns. Figure 5a clearly shows the prominent graphene-like flake structure of the LIG pattern. Most of the structure, except for residual areas of the polyimide film, is carbonized to the bottom.

The chemical structure of the LIG pattern was further investigated using a micro Raman spectrometer (NRS-5100). As can be seen in Figure 5b, the Raman spectrum of the LIG pattern clearly shows three distinct peaks: the D peak at 1350 cm^−1^ was induced by sp2 carbon bonds; the G peak at 1580 cm^−1^ and the 2D peak at 2700 cm^−1^ show similar shapes, proof of graphene’s graphene-like flake structure and confirmation of the results of previous studies on LIG [18,19,20,21,22,23,24,30,31,32,33]. The analysis of the 2D peak provides important information for estimating the number of layers and the stacking order in graphene or graphic layers [30]. LIG patterns have been successfully implanted into PDMS by adjusting the laser fluence through changing various laser settings such as laser power and scanning speed.

### 3.2. Piezoresistive Effect of Fabricated Strain Sensor

A porous and graphene-like flake-structured LIG pattern was implanted on a flexible PDMS substrate and then protected again with PDMS to produce a bendable and highly sensitive strain sensor (Figure 6).

In order to evaluate the sensing properties of the strain sensors, we studied the strain-induced changes in electrical resistance of the LIG patterns. As shown in Figure 7a, when the strain sensor is fabricated by irradiation with a 355 nm pulsed laser with a laser fluence of 7.4 J/cm^2^, the electrical resistance to strain changes from 234 to 837 Ω, which means that the porous LIG pattern shows high sensitivity (GF ~ 160) even for fine strain (1.6%). When the bending distance changed, the corresponding variation in the electrical resistance increased linearly and the GF values of the laser fluence at 6.4, 6.1, and 5.6 J/cm^2^ were 8.5, 4.18, and 2.03, respectively (Figure 7b).

As can be seen in Figure 7b, when the laser fluence is higher, the polyimide film is carbonized more by photothermal ablation and has better electrical properties and higher sensitivity as a strain sensor. Moreover, when the sensor is fabricated with a high laser fluence, the depth of the graphitic pattern of the polyimide film becomes deeper, the pattern is better embedded in the substrate, and the durability of the sensor becomes stronger. As shown in Figure 8, fracture of the strain sensor was caused by 30% strain and was caused entirely by the fracturing of the PDMS. 

Cyclic tensile testing was applied to the strain sensor; we evaluated the piezoresistive effect by measuring the electrical resistance with a sourcemeter (KEITHLEY, sourcemeter 2450). As can be seen in Figure 9, it is possible to evaluate the response of the electrical resistance when the sensor is periodically stretched/released at the same strain (ε = 1%, 3%, and 7%). Even after the cycle test (over 200 iterations), the sensor demonstrated no hysteresis. All sensors increase–decrease the electrical resistance as strains increase–decrease accordingly. In addition, for all applied strains, the initial resistance of the LIG strain sensor did not change after 200 cycles of testing. Since our LIG strain sensor had its LIG pattern embedded in PDMS, sensor linearity (R^2^ = 0.9984) was far better than those of the previously reported silver nanowire sensor (R^2^ = 0.94), CNT-based sensor (nonlinear), and graphene-based strain sensors (nonlinear) [4,5,6,7,8,9].

As shown in the experimental results, our strain sensors can be applied in many areas with advantages such as high sensitivity, fast response speed, high linearity, and low hysteresis. Figure 10a shows the current–voltage characteristics according to the different strain conditions of the LIG strain sensor and indicates that Ohm’s law is followed linearly when tensile strain is applied. When a certain amount of voltage is applied to the sensor, it is confirmed that the electrical resistance increases as the strain of the sensor increases and, therefore, the current value becomes smaller. LIG strain sensors have been fabricated using different focal lengths; for the resulting GFs, initial electrical and sheet resistances are shown in Figure 10b. The fabricated LIG strain sensor showed higher GF, and lower electrical resistance and sheet resistance as the laser focal length was matched to that of polyimide. This indicates that the degree of carbonization of the polyimide film has a great influence on the LIG sensor. 

In summary, the sensitivity, linearity, and elasticity of a sensor can be adjusted according to the degree of carbonization and the embedding of the pattern; parameters can be selected according to the application field. LIG strain sensors with low electrical resistance are suitable for applications requiring high GF and high strain. On the other hand, low electrical resistance strain sensors can be used in applications requiring high linearity. The response time for strain sensing was about ~70 ms, measured with a data acquisition (DAQ) system; the delay time due to bending was negligibly small (Appendix A). This is due to van der Waals forces between the carbon network [14,17] of the LIG pattern and the PDMS used as the top layer. The electrical properties of our sensors follow Ohm’s law in a very linear manner.

### 3.3. Application of the LIG Strain Sensor

Due to the flexible and highly sensitive nature of the LIG strain sensor, application to human motion monitoring was considered. In terms of potential evaluation of our sensor’s application to wearable devices, we built a glove sensor that combined latex gloves and sensors. As shown in Figure 11, our sensor was fully responsive to finger movements. We strained the sensor by bending the middle finger joint to a specific angle. As the finger bends more and more, i.e., as the bending angle increases, the strain applied to the sensor increases, thereby increasing the electrical resistance. Also, when we straightened the middle finger again, the sensor returned to its initial electrical resistance value. Moreover, the LIG strain sensor showed very good durability because it returned to its initial state even after repeated finger-bending cycle experiments.

## 4. Conclusions

We report, in this paper, highly sensitive and flexible piezoresistive type strain sensors and demonstrate their valuable use in strain sensing in the motion-monitoring industry. The LIG strain sensor was investigated and found to have a response time of ~70 ms, good linearity to tensile strain, high GFs that react to bending, and a good degree of creep in the bending/release cycle. There was no fracturing due to bending. Also, due to its powerful piezoresistive effect, the sensor has a GF range of 1~160; its stretchability can reach ~30% depending on the carbonization degree of the LIG pattern, and PDMS was destroyed after strain of 30%. Laser power, scanning speed, and focal length were the main parameters involved in the degree of carbonization of the pattern. The LIG pattern shows various electrical characteristics such as electrical resistance and sheet resistance; stretchability was also demonstrated in various ways. Finally, we investigated the possibility that the LIG strain sensor could be applied in industry for human motion detection. We attached the sensor to a latex glove and made a glove sensor. Finger joint bending was monitored by our glove sensor made with the LIG strain sensor.

## Figures and Tables

**Figure 1 sensors-19-04867-f001:**
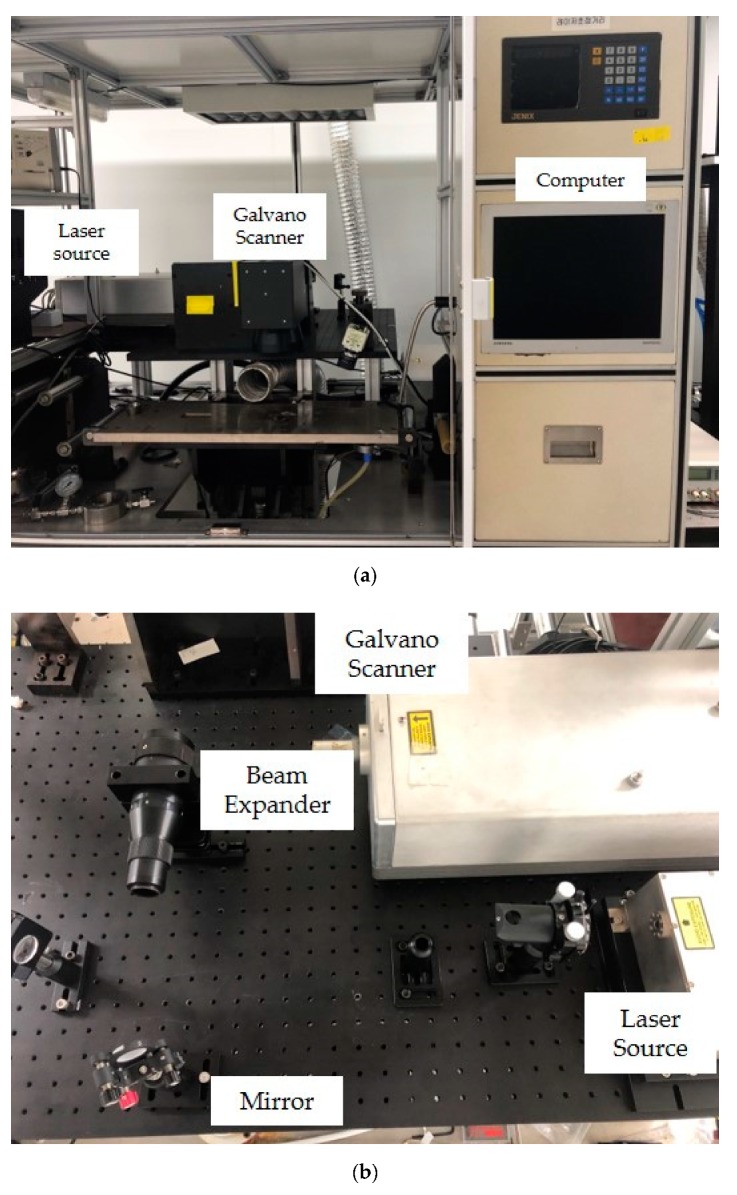
Laser system: (**a**) laser setup and (**b**) optical system.

**Figure 2 sensors-19-04867-f002:**
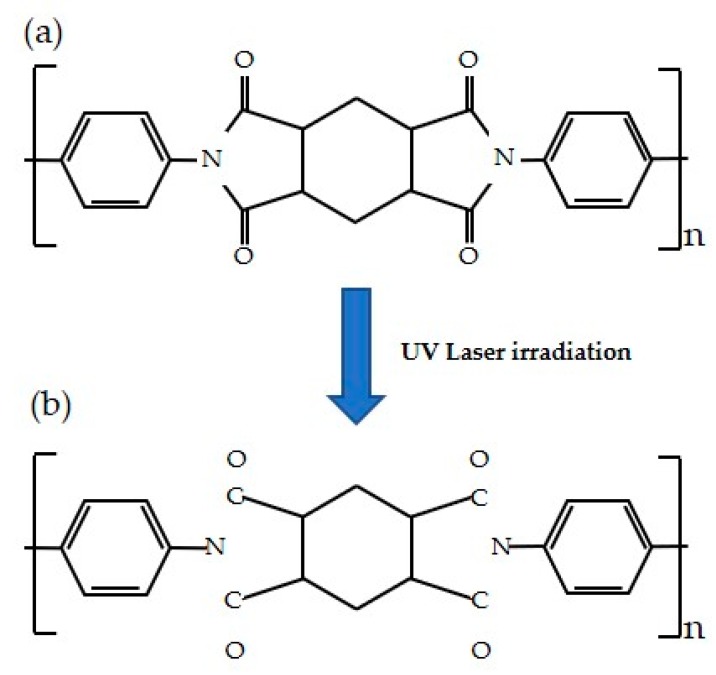
Principle of laser-induced graphene (LIG). (**a**) Structure of polyimide film; (**b**) Structural change of polyimide film after laser irradiation.

**Figure 3 sensors-19-04867-f003:**
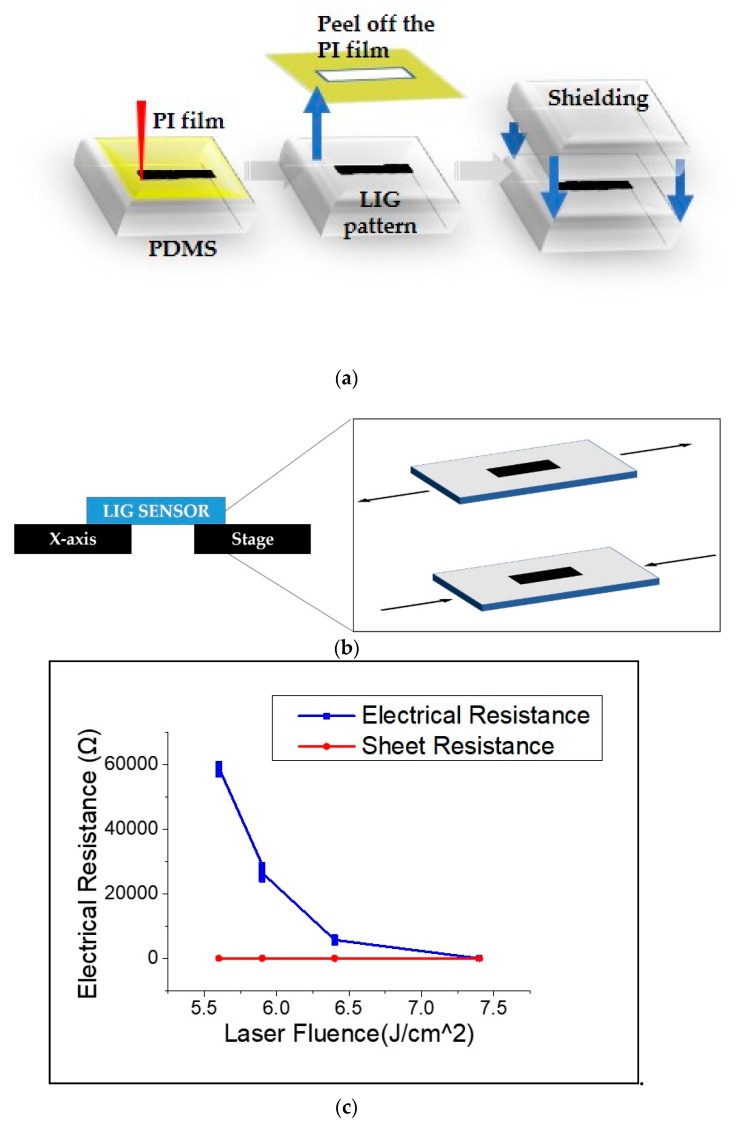
Fabrication of LIG sensor: (**a**) process of fabricating the LIG strain sensor, (**b**) strain generation by bending, and (**c**) changes in electrical characteristics according to laser fluence.

**Figure 4 sensors-19-04867-f004:**
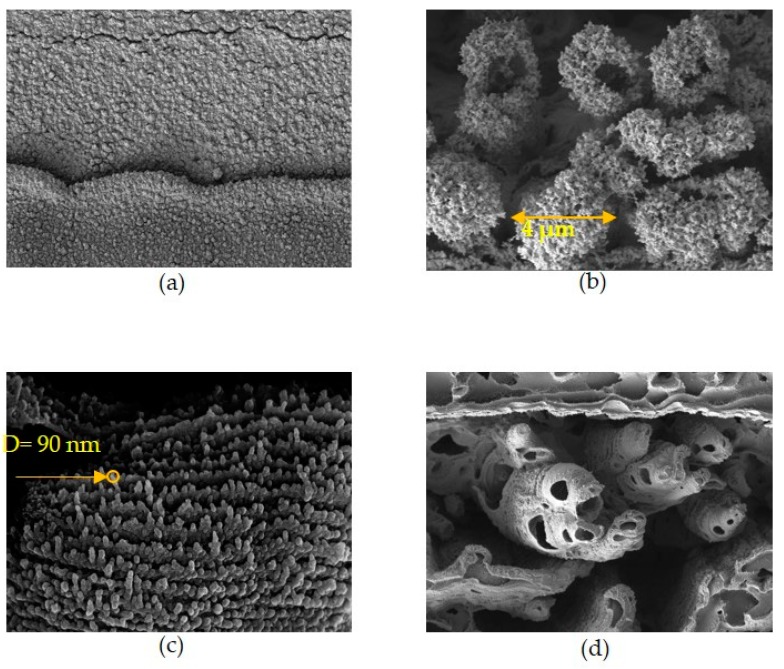
Laser fluence-dependent SEM images of LIG pattern; laser fluence of (**a**) 5.6 J/cm^2^; (**b**) 5.9 J/cm^2^; (**c**) 6.4 J/cm^2^; (**d**) 7.4 J/cm^2^.

**Figure 5 sensors-19-04867-f005:**
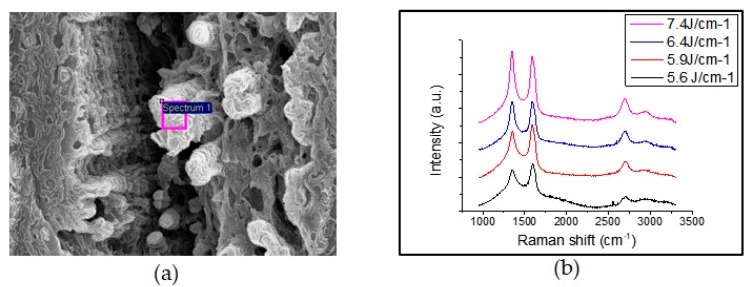
The graphene-like flake structure of the LIG pattern; (**a**) SEM image of the graphene-like flake structure of LIG; (**b**) Raman spectra of the LIG pattern irradiated laser with different laser fluences.

**Figure 6 sensors-19-04867-f006:**
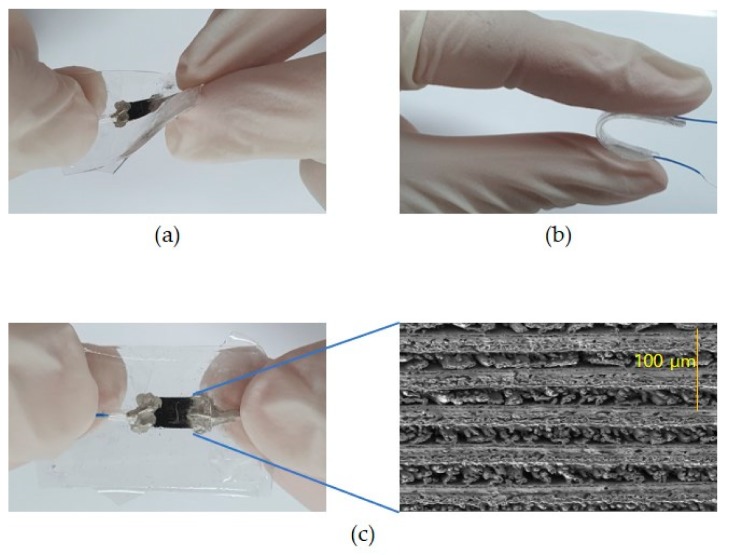
Photographs of the strain sensor under (**a**) twisting and (**b**) bending; (**c**) SEM image of carbonized pattern embedded on polydimethylsiloxane (PDMS).

**Figure 7 sensors-19-04867-f007:**
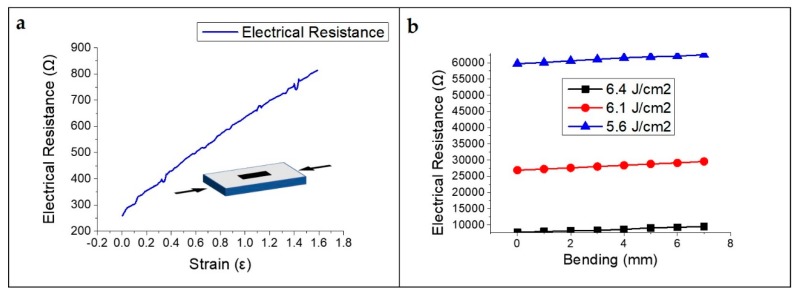
Change in electrical resistance corresponding to strain with each laser fluence: (**a**) 7.4 J/cm^2^; (**b**) 5.6 to 6.4 J/cm^2^.

**Figure 8 sensors-19-04867-f008:**
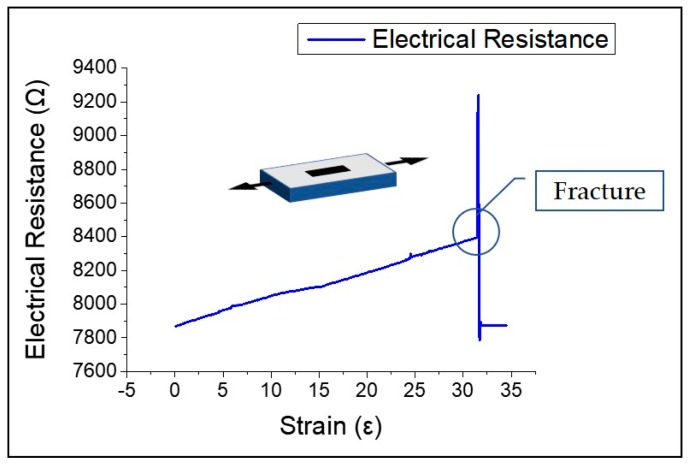
Change in electrical resistance corresponding to strain and fracture.

**Figure 9 sensors-19-04867-f009:**
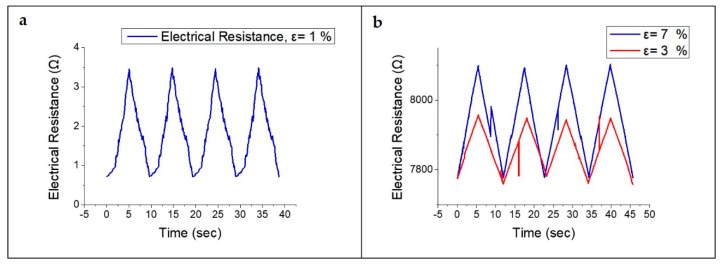
Response of LIG strain sensor under stretch/release cycle of (**a**) ε = 1%; (**b**) ε = 3% and 7%.

**Figure 10 sensors-19-04867-f010:**
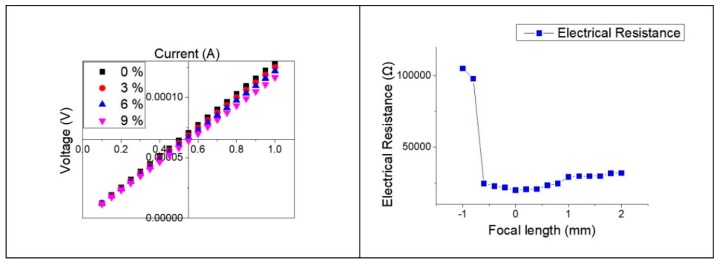
(**a**) Electrical response of LIG strain sensor; (**b**) Change of initial electrical resistance according to focal length.

**Figure 11 sensors-19-04867-f011:**
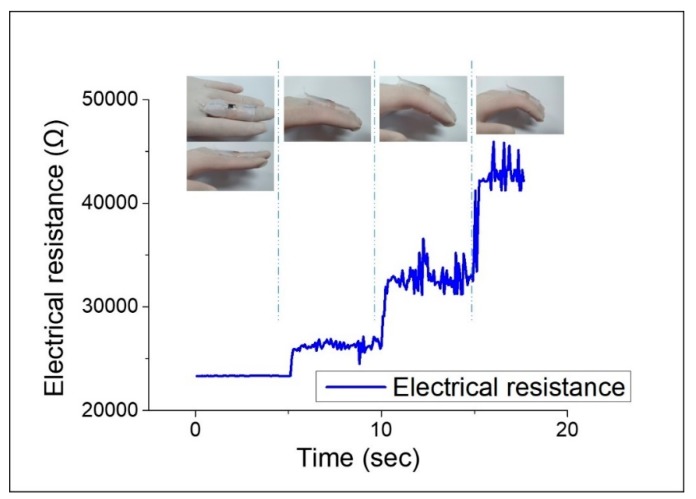
Motion detection of middle finger.

**Table 1 sensors-19-04867-t001:** 355 nm UV pulsed laser specifications.

Parameter	Unit	Value
Wavelength	nm	355
Average power	Watt	~2.5
Pulse length	ns	25
Repetition rate	kHz	30
Mode		TEM_00_
Beam diameter	mm	0.4

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
