# Peer review of "Flexible and Highly Sensitive Strain Sensor Based on Laser-Induced Graphene Pattern Fabricated by 355 nm Pulsed Laser"

_sensors, 2019, doi:10.3390/s19224867_

Round 1
Reviewer 1 Report
This manuscript reports on the Laser Induced carbonization of Polyimide for the implantation of multilayer graphene patterns on PDMS. The proposed application is a strain sensor with GF up to 160.
The manuscript is well structured with novel features highlighted. Some issues remain unclear and should be addressed before the manuscript is accepted for publication.
An estimation of thickness of laser induced graphene correlated to the number of layers. The presented graphene structure resembles porous multilayer graphene. Also, Raman spectra match well reduced Graphene Oxide or multilayer graphene owing to the d-peak intensity and the high g/2d ratio. Do the authors have an estimation of the number of layers? In previous works, laser induced transfer of highly transparent networks of nanowires or thin films on polymers such as PDMS, has been reported. Please cite some of these works and highlight the difference of your method with laser induced transfer carried out in contact with a polymer receiver substrate Reference 30 is missing and is necessary to support the authors’ claim.
Author Response
Hello, This is Jeong.
First of all, thank you for your good review. Your advice will improve my paper.
I transfer the term 'multi-layer' to 'graphene-like flake'.
Because I think I can't estimate accurate numbers of the layer of the graphitic pattern. The reason why I can't estimate this is I fabricated the graphene-like layer on a polyimide film. In my opinion, carbon atoms have been created in carbon-based substrates(polyimide film), so even if I estimate the number of layers, the error will be too large.
And for comparison of Raman spectrum, 30-32 papers have been added to the reference.
Reviewer 2 Report
This paper has many basic errors and shows no relative proficient in the strain sensor. Furthermore, that shows less new ideas about designing and engineering the strain sensor. Here comments to show that there is much room to improve this paper.
Firstly, in Figure 2a, the formula of the polymers should mark its degree of polymerization(DOP). If DOP of the chemicals cannot be determined, it should be given “n” writing down the right-bottom corner of the bracket of the formula. In the line of 111, there is no Figure 1c in this paper, but the authors try to describe its feature in this paper. Continuously, in the line of 105, the authors gave the formula calculating the gauge factor, but there is no G.F which can be found in this paper. The acronym of the gauge factor is “G.F.” instead of “G.F”. Figure 3 has two “a” to mark the first sub-figure. It involves plagiarism to explain the mechanism of the strain generation by bending in Figure 3b. Here I can put on the website of this picture
(https://www.doitpoms.ac.uk/tlplib/beam_bending/printall.php). By the way, the authors do not choose a uniform style to illustrate the figures throughout the whole passages. For example, In the line of 93, they chose ‘Figure 3a’ as the caption, but they chose “Fig. 3b” as the caption in the line of 109. Next, not only do the Figure 4 includes the porous LIG patterns, but also it contains the Raman shift of the patterns. However, the authors only mentioned the part of the information in the line of 131 and 141. The “GF” has been defined in the line of 102, but there is a definition of “GF” in the line of 169, repetitively. Figure 7 has provided no signs to mark its sub-figures(“a” or “b”), and Figure 7, furthermore, fails to present the information about the current-voltage characteristics in the line of 204. Also, the paper never mentions in Figure 8. Essentially, Figure 8 presents information about the current-voltage plot.
Moving to the aspect of the investigation, Figure 3b shows that the angle(theta) equals “1” over “R”, but the angle is dimensionless. The dimension of “1” over “R” is reciprocal length. This expression is wrong because its dimension of the left-hand does not match its dimension of the right-hand. As previously stated, this picture involves plagiarism. In the original text, its author explained that the unit of “1”, and however this paper has no explanation about that. In other words, the formula presented in Figure 3b can only apply for the small strain rather than that Figure 5b shows the finite large deformation. Additionally, Figure 5c presents the laminated structure of the carbonized pattern, it is more difficult to describe the plate with laminated structure than do researchers describe the single-layer plate. In J.G.Ren’s paper(Composites Science and Technology 27 (1986) 225-248) might provide some helpful knowledge about the laminated plate. And then, Figure 3c and Figure 6b shows the tendency of the changes of the electrical resistance, whereas the 150 line states that “… the stronger the laser fluence is, the more porous the structure is… “. Those two results seems to be contradictory. The more porous the structure is, the larger the permittivity the structure is, so the electrical resistance would increase. In the line of 180, this paper presents no evidence to prove that the strain sensor of this work fractured under bearing 30% strain. Next, there is no evidence to show that the strain sensor experienced the test of 200 times in the line of 194. Moving to the next line, the paper should list the key results from reference[4-9] in detail. This paper presents the cycling tests under strain 1, 3, and 7 percent, except 5 percent. Because of this, are there any special reasons to ignore the condition of 5 percent? And the same problem is that there is no data to support the sensor which can response applied strain immediately with about 70ms in the line of 216. Moving to Figure 9, while the strain sensor binding the middle finger bends in the stage of large deformation, its electrical resistance fluctuates widely and sharply.
To recapitulate, this paper has many basic errors about marking the figures firstly, and secondly, it provides some conclusions but the authors didn’t offer the relative evidence. What’s more, the paper also shows few advantages than that other researchers performed works about strain sensors(piezoresistive sensor). The last, authors might avoid using any individual’s picture without authority in academical writing.
Author Response
First of all, thank you for your full review.
Your review will make my paper better.
1. I revised all the errors regarding the figures you mentioned.
2. Added a graph that breaks the sensor at more than 30% strain (Figure 10).
3. In the case of a cycle test, the reason why the 5% strain is empty is that since the bending was applied to the sensor at a certain distance, it would go straight from 3% to 7% strain. (ex, 1mm, 5mm, 10mm)
4. The response time of 70ms of the sensor was measured by DAQ, and the video will be attached.
Reviewer 3 Report
This manuscript discusses the development and initial testing of a LIG pattern based strain sensor. It looks interesting. However, the manuscript needs major changes to enhance the quality . Some of my comments are below:
Abstract needs restructuring: better if first sentence (line 10-11) is re-phrased. Line 14: Focal length related results are not given/clear. Line 16: 'fracture at 30% tensile strain' .... no proof for this.
Referencing errors throughout the manuscript. eg: line 25, a dot appears before the references '.[1-15].' This should be '[1-15].'
Other errors:
Line 22: PEI ... spelling error (an additional 'e' appears)
Line 60: expand DLW
line 66-67: Says 'pulsed laser to form LIG'. No explanation of LIG making. Add a reference. Include details of the polyimide film properties... its fabrication/making, thickness, substrate, etc.
line 71 (onwards): referencing style changed.
72: 1000 'C ? ... is this degrees?
74: 'fig 2 shows the generation principle of the LIG'... Explanation required for the chemical changes. Add references to justify this.
79: better if on of the pictures is replaced with a schematic of the experimental setup. Sample position should be shown.
101: 'stetchability'.... spelling error?
109: Fig. 3b is not explained.
111: Fig 1c not found!
122 : J/cm^2 ... not J/cm2
135: all the images should have the scale. Currently, not comparable! Also, discuss in detail about the changes, reasons,.. Also, link the Raman spectra.
a rectangle in fig 4e.. why? what was the fluence?
153: include the laser wavelength of the Raman system.
165-66: describe the expanded figure showing layers.. what are these layers?
170-172: Rephrase the sentence.
172: 'x-direction bending' ... use properly. why x-direction not y?
175: Fig 5b... not the right figure!
180-181: what is the proof (results) for the statement 'fracture at 30%% strain'?
183: in figure 6b, 'bending distance' .. not clear. use appropriate word.
193: 'did not change after 200 cycles' ... any proof for this?
204: figure 7a?
206-207: rephrase the sentence. ... 'fabricated using different focal lengths'?
where is fig. 7b?
218-219: 'sensors follow Ohm's law'.. not clear. Rephrase.
237: No proof for ~70 ms response time!
245: 'applied to industry' ... better if 'applied in industry'
Author Response
First of all, thanks to your high-level review, I appreciate that my paper can be further advanced.
My answer to your review is as follows:
1.?In the abstract, we wrote about the focal length (line14) and explained that we found the optimal focal length to produce the LIG from the results of the paper and graphs.
2. Line 60:?The term Direct Laser Writing (DLW) was used in 56 lines.
3. Line 66-67:?The theoretical part of LIG was described in line 28-35 with references.
4. Line 74:?A description of Figure 2 has been added to Line 75-77, and Reference 33 has been added.
5. Line 109:?The figure has been modified.
6. Line 135:?The analysis of Raman Spectra was detailed in lines 158 to 166.
7. Figure 6c: The figure is a surface SEM image of the LIG pattern, not a layer.
8. Line 180: This is not an X-axis or Y-axis, but a one way bending of the sensor. Therefore, the expression x-direction has been deleted.
9. There is a term ; Stretchability in the dictionary.
(https://www.dictionary.com/browse/stretchability)
10. We use a device called DAQ to measure the response of the sensor.
(To prove this, I attach two videos.)
Round 2
Reviewer 2 Report
After reviewing the revised version, I am fine with it and recommend its publication.
Reviewer 3 Report
It is a reasonably good looking paper.